# Impact of universal varicella vaccination on the use and cost of antibiotics and antivirals for varicella management in the United States

Manjiri Pawaskar[1]*, Jaime Fergie[2], Carolyn Harley[3], Salome Samant[1], Phani Veeranki[3], Oliver Diaz[3], James H. Conway[4]

1 Merck & Co. Inc., Rahway, New Jersey, United States of America, 2 Driscoll Children's Hospital, Corpus Christi, Texas, United States of America, 3 PRECISIONheor, Los Angeles, California, United States of America, 4 School of Medicine and Public Health, University of Wisconsin, Madison, Wisconsin, United States of America

* manjiri.pawaskar@merck.com

## Abstract

### Background

Our objective was to estimate the impact of universal varicella vaccination (UVV) on the use and costs of antibiotics and antivirals for the management of varicella among children in the United States (US).

### Methods

A decision tree model of varicella vaccination, infections and treatment decisions was developed. Results were extrapolated to the 2017 population of 73.5 million US children. Model parameters were populated from published sources. Treatment decisions were derived from a survey of health care professionals' recommendations. The base case modelled current vaccination coverage rates in the US with additional scenarios analyses conducted for 0%, 20%, and 80% coverage and did not account for herd immunity benefits.

### Results

Our model estimated that 551,434 varicella cases occurred annually among children ≤ 18 years in 2017. Antivirals or antibiotics were prescribed in 23.9% of cases, with unvaccinated children receiving the majority for base case. The annual cost for varicella antiviral and antibiotic treatment was approximately $14 million ($26 per case), with cases with no complications accounting for $12 million. Compared with the no vaccination scenario, the current vaccination rates resulted in savings of $181 million (94.7%) for antivirals and $78 million (95.0%) for antibiotics annually. Scenario analyses showed that higher vaccination coverage (from 0% to 80%) resulted in reduced annual expenditures for antivirals (from $191 million to $41 million), and antibiotics ($82 million to $17 million).

**Data Availability Statement:** All relevant data are within the paper and its Supporting information files.

**Funding:** This study was sponsored by Merck Sharp & Dohme LLC, a subsidiary of Merck & Co., Inc., Rahway, NJ, USA (MSD). The funder had no role in study design, data collection and analysis, decision to publish, or preparation of the manuscript. M. Pawaskar and S. Samant, are employees of MSD and own stocks in Merck & Co., Inc., Rahway, NJ, USA. P. Veeranki, and C. Harley, are employees of PRECISIONheor, which received financial support from MSD, for the execution of this research. Though they received no payment for their work on this study, J. H. Conway reports grants and personal fees from Sanofi Pasteur, Pfizer, Merck, GSK, and Centers for Disease Control outside of the submitted work while J. Fergie reports personal fees from MSD, outside the submitted work.

**Competing interests:** M. Pawaskar and S. Samant, are employees of Merck Sharp & Dohme Corp., a subsidiary of Merck & Co., Inc., Kenilworth, NJ, USA, and own stocks in Merck & Co., Inc., Kenilworth, 257 NJ, USA. P. Veeranki, and C. Harley, are employees of PRECISIONheor, which received financial support from Merck Sharp & Dohme Corp., a subsidiary of Merck & Co., Inc., Kenilworth, NJ, USA for the execution of this research. Though they received no payment for their work on this study, J. H. Conway reports grants and personal fees from Sanofi Pasteur, Pfizer, Merck, GSK, and Centers for Disease Control outside of the submitted work while J. Fergie reports personal fees from Merck Sharp & Dohme Corp, outside the submitted work.

## Conclusions

UVV was associated with significant reductions in the use of antibiotics and antivirals and their associated costs in the US. Higher vaccination coverage was associated with lower use and costs of antibiotics and antivirals for varicella management.

## Introduction

Varicella (chickenpox) is a highly contagious, acute infectious disease caused by the varicella zoster virus [1–3]. Though chickenpox typically presents in early childhood, (often <10 years of age in temperate countries), it can also occur in older children and adults [1–3]. It usually presents with fever, malaise and skin rash characterized by generalized, pruritic vesicles in crops [2,3]. While varicella is often mild and self-limiting, serious complications including secondary bacterial skin infections, bacterial and viral pneumonia, and cerebellar ataxia can occur, resulting in hospitalization and rarely death [2–4]. Secondary bacterial infection with *Staphylococci* or *Streptococci* is the most common complication in children, especially infants [2,3]. Risk of complications is higher in immunocompromised persons, neonates and adults [3,5].

The treatment of varicella and its complications depends on age, severity and underlying health status [6]. Most cases can be treated with over-the-counter medications. Antiviral medications such as acyclovir are not recommended for healthy children < 12 years, but may be considered in patients at high risk for moderate or severe varicella infection [7]. Antibiotics are not typically indicated for the management of primary varicella infection, but may be prescribed to treat complications associated with varicella infection such as secondary bacterial skin infections, bacterial pneumonia or other bacterial complications. A multi-country study of the burden of varicella reported that antibiotics were prescribed to nearly 13% of outpatient varicella cases and almost 70% of inpatient cases [8].

Vaccination has proven to be highly effective at reducing varicella-related morbidity and mortality [2–4,9]. The US was the first country to implement universal varicella vaccination (UVV) following Food and Drug Administration approval in 1995 (VARIVAX®, Varicella Virus Vaccine Live, Merck & Co., Inc., Kenilworth, NJ, USA) [10]. A second dose for all recipients was approved in 2005 and recommended by the Advisory Committee on Immunization Practices (ACIP) in 2006 [3,10,11]. Between 1995 and 2010, annual varicella cases in the U.S. decreased by 92%, hospitalizations by 84%, and varicella-related deaths by 90% [12]. Previous economic analyses have suggested that UVV programs offer substantial financial savings as well as societal benefits, largely derived from direct and indirect costs related to varicella prevention. These savings are consistent regardless of the country studied and the type of healthcare system in place [13,14].

In spite of the proven benefits of UVV, there are only 39 countries that have included varicella vaccination in their national immunization program [8,10]. Considering the on-going challenges with antibiotic resistance globally, this study aimed to estimate the potential reduction in the annual burden of antibiotic and antiviral use in the US due to the universal varicella vaccination program [9,15,16].

## Methods

We developed a decision tree model to estimate the annual number of varicella cases, antiviral and antibiotic prescriptions, and costs among children aged < 18 years in the US in 2017

under different vaccination coverage scenarios. (Fig 1). Model inputs included the proportion of children vaccinated against varicella (unvaccinated, one dose and 2 doses), annual varicella incidence by vaccination status, the proportion of children who developed varicella with and without complications by vaccination status, the proportion prescribed antivirals and antibiotics for varicella with and without complications, and associated costs [17,18]. The inputs were uniform across ages and did not account for herd immunity benefits. Inputs were obtained from published sources with details provided in Tables A-D in S1 Appendix.

Treatment decision information (i.e. the probability of antibiotic therapy, antiviral therapy, both, or none for each varicella case) was based on a 2019 survey of health care providers who provided their electronic written consent, in a study which was determined to be exempt from Institutional Review Board (IRB) oversight by Advarra IRB (Table E in S1 Appendix) [19]. As the model used secondary, published data, additional IRB review was not required. Antibiotic and antiviral drug costs were calculated for the most commonly prescribed antibiotics and antivirals for varicella for children from published sources and expert opinion (Tables C-D in S1 Appendix). The base case and scenarios used probabilistic sampling and 10 million replications to produce estimates of antibiotic and antiviral medication prescriptions and related costs.

In the base case, outcomes were estimated over one year using the vaccination coverage rate in the U.S. (93.8% receiving 2 doses, 2.4% receiving a single dose, and 3.8% unvaccinated against varicella) [17]. We evaluated additional scenarios that potentially reflect varicella vaccination globally. Scenarios modeled were: no vaccination (i.e., 0% vaccinated) reflecting countries without any vaccination programs; 20% vaccinated, reflecting countries who have varicella vaccination available through private insurance but no UVV; and 80% vaccinated

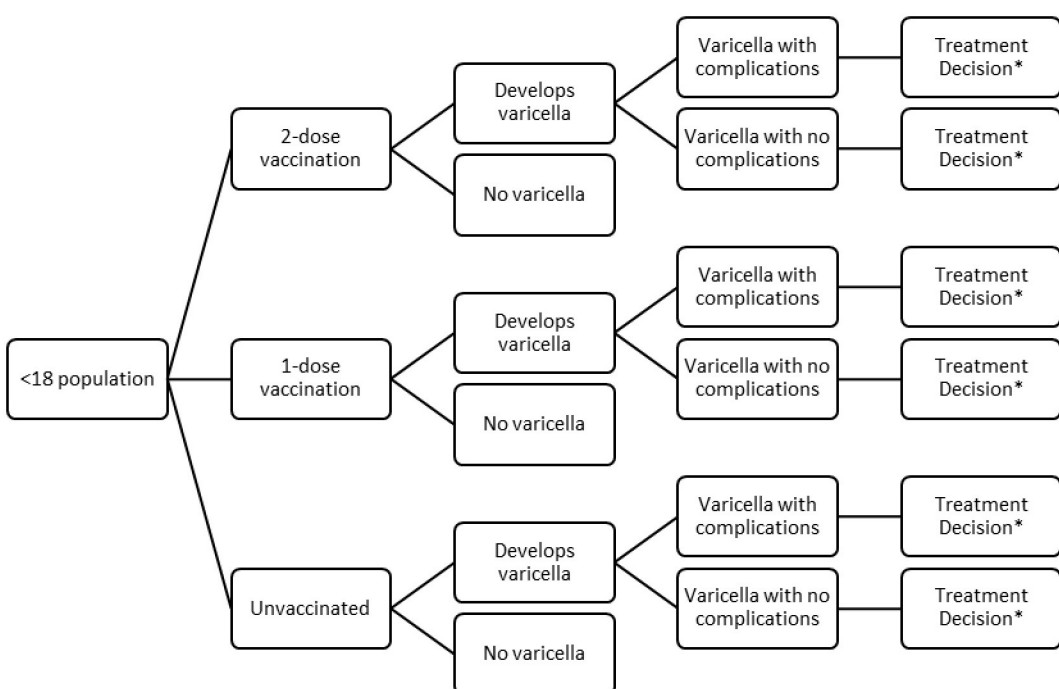

**Fig 1. Varicella model structure.** Varicella Model Structure: This figure represents the vaccination states of patients and possible treatment decisions within our model. * * *Treatment decisions could be antibiotic therapy, antiviral therapy, both, or none for each varicella case.

(the minimum vaccination coverage, recommended by WHO for implementing UVV.) [2]. To estimate annual prescriptions and costs in US dollars for 2020 prevented through the UVV program in the US, the base case was compared to no vaccination scenario. Costs were reported in 2020 USD.

## Results

### Varicella cases

Results of the base case model are presented in Table 1. Under current vaccination coverage rates, a total of 551,434 cases of varicella were estimated to occur annually among US children < 18 years of age with 396,633 varicella cases among unvaccinated children, 14,406 cases among those vaccinated with a single dose and 140,395 cases among those vaccinated with 2 doses of varicella vaccine (See Table A in S1 Appendix for details). The majority of cases occurred in unvaccinated children (71.9%, 396,633 cases); 94.5% (521,143 cases) of cases had no complications while 5.5% (30,291) of all cases had some complication.

### Antibiotics and antiviral prescription use

The model estimated that antivirals or antibiotics were prescribed in 23.9% of all varicella cases annually (Table 1). Unvaccinated children received the majority (73.7%) of total prescriptions, accounting for 72.0% and 77.5% of all antivirals and antibiotics prescribed, respectively (Table H in S1 Appendix). Cases with complications were more likely to be prescribed medications (91.4%) compared to varicella cases without any complications (21.5%) (Table F in S1 Appendix). For cases with complications, antibiotics (68.7% of all prescriptions) were more commonly prescribed than antivirals (31.3%) while the reverse was true for cases with no complications (75.2% antivirals versus 24.8% antibiotics) (Table F in S1 Appendix).

**Table 1. Estimated annual varicella cases, antibiotic prescriptions and antiviral prescriptions in the US population under 18 years old by vaccination status, base case.**

|  | Unvaccinated | 1-Dose (partial) | 2-Dose (full) | Total |
|---|---|---|---|---|
| Population | 2,796,155 (3.8%) | 1,765,992 (2.4%) | 69,020,868 (93.8%) | 73,583,015 (100%) |
| Varicella cases (% of population) (Row %) | 396,633 (14.2%) (71.9%) | 14,406 (0.8%) (2.6%) | 140,395 (0.2%) (25.5%) | 551,434 (0.7%) (100%) |
| Cases with no complications (% of Cases) | 369,625 (93.2%) | 14,029 (97.4%) | 137,490 (97.9%) | 521,143 (94.5%) |
| Cases with complications (% of Cases) | 27,007 (6.8%) | 378 (2.6%) | 2,906 (2.1%) | 30,291 (5.5%) |
| Cases with antiviral or antibiotic prescription (% of cases) | 96,812 (24.4%) | 3,276 (22.7%) | 31,667 (22.6%) | 131,754 (23.9%) |
| Antiviral prescription only (% of cases) (row %) | 66,988 (16.9%) (72.0%) | 2,428 (16.9%) (2.6%) | 23,681 (16.9%) (25.4%) | 93,098 (16.9%) (100%) |
| Antibiotic prescription only (% of cases) (row %) | 33,766 (8.5%) (77.5%) | 955 (6.6%) (2.2%) | 8,869 (6.3%) (20.3%) | 43,590 (7.9%) (100%) |
| Total prescriptions (Row %) | 100,755 (73.8%) | 3,383 (2.5%) | 32,550 (23.8%) | 136,688 (100%) |

Base case scenario. Results were extrapolated to the 2017 population of 73.5 million US children (0–18 years) with vaccination coverage as follows: Unvaccinated: 3.8%; 1 dose: 2.4%; 2 doses: 93.8%. See supplement for details.

## Medication costs

Under the base case scenario, the current vaccination rates resulted in savings of $181 million (94.7%) for antivirals and $78 million (95.0%) for antibiotics annually compared to no vaccination. The annual cost of antivirals and antibiotics associated with varicella was estimated to be $14 million ($26 per case), with varicella cases with no complications accounting for nearly $12 million ($23 per case), and those associated with complications accounting for just over $2 million ($71 per case) (Table 2; Table G in S1 Appendix). Unvaccinated children accounted for 73.5% of total costs, including $7.3 million in antiviral costs and $3.1 million in antibiotic costs annually.

## Scenario analysis

Results from the three modeled scenarios (no vaccination, 20% vaccinated, and 80% vaccinated) are presented in Figs 2, 3 and 4. The total number of varicella cases in each scenario decreased as vaccination coverage increased, from 10,437,699 (no vaccination scenario) to 8,382,342 (20% vaccinated) to 2,216,273 cases (80% vaccinated scenario) (details provided in Table H in S1 Appendix). A similar pattern was evident for the number of antivirals and antibiotics prescribed, with the no vaccination scenario having the most antiviral and antibiotic prescriptions (1,762,865 annual antiviral and 888,581 annual antibiotic prescriptions), and the 80% vaccination scenario having the least (374,285 annual antiviral and 185,886 annual antibiotic prescriptions). Annual antiviral and antibiotic prescriptions ranged from $41 to $191 million in drug costs while antibiotics ranged from for $17 to $82 million, depending on scenario (Fig 4) (see Table I and L in S1 Appendix for details).

Compared to no vaccination, the current UVV program in the US resulted in an estimated reduction of the use of antiviral and antibiotic prescriptions of 94.6% annually (2,020,361 fewer total annual prescriptions) for varicella cases with no complications and by 95.7% (494,397 fewer total annual prescriptions) for varicella with complications (Table 2; Tables J and K in S1 Appendix). This resulted in a 94.8% reduction ($259,064,629) in antibiotic and antiviral costs compared to the no vaccination scenario (Table L in S1 Appendix).

**Table 2. Estimated total number of prescriptions and associated costs of antiviral and antibiotic use for the base case (Varicella cases with and without complications).**

|  | Antivirals | Antibiotics | Total |
|---|---|---|---|
| Annual prescription (n) for |  |  |  |
| Cases with no complications | 86,188 | 28,427 | 114,615 |
| Cases with complications | 6,910 | 15,164 | 22,074 |
| Total | 93,098 | 43,591 | 136,688 |
| Annual costs ($) for |  |  |  |
| Cases with no complications | $9,335,884 | $2,630,542 | $11,966,426 |
| Cases with complications | $748,491 | $1,403,277 | $2,151,768 |
| Total | $10,084,375 | $4,033,819 | $14,118,194 |

*Base case: Current vaccination scenario in USA with unvaccinated: 3.8%; 1 dose: 2.4%; 2 doses: 93.8%.

** Compared to No vaccination scenario, the Base case scenario had 95.9% fewer annual prescriptions (153,877 fewer antiviral and 340,521 fewer antibiotic prescriptions); See supplement for further details.

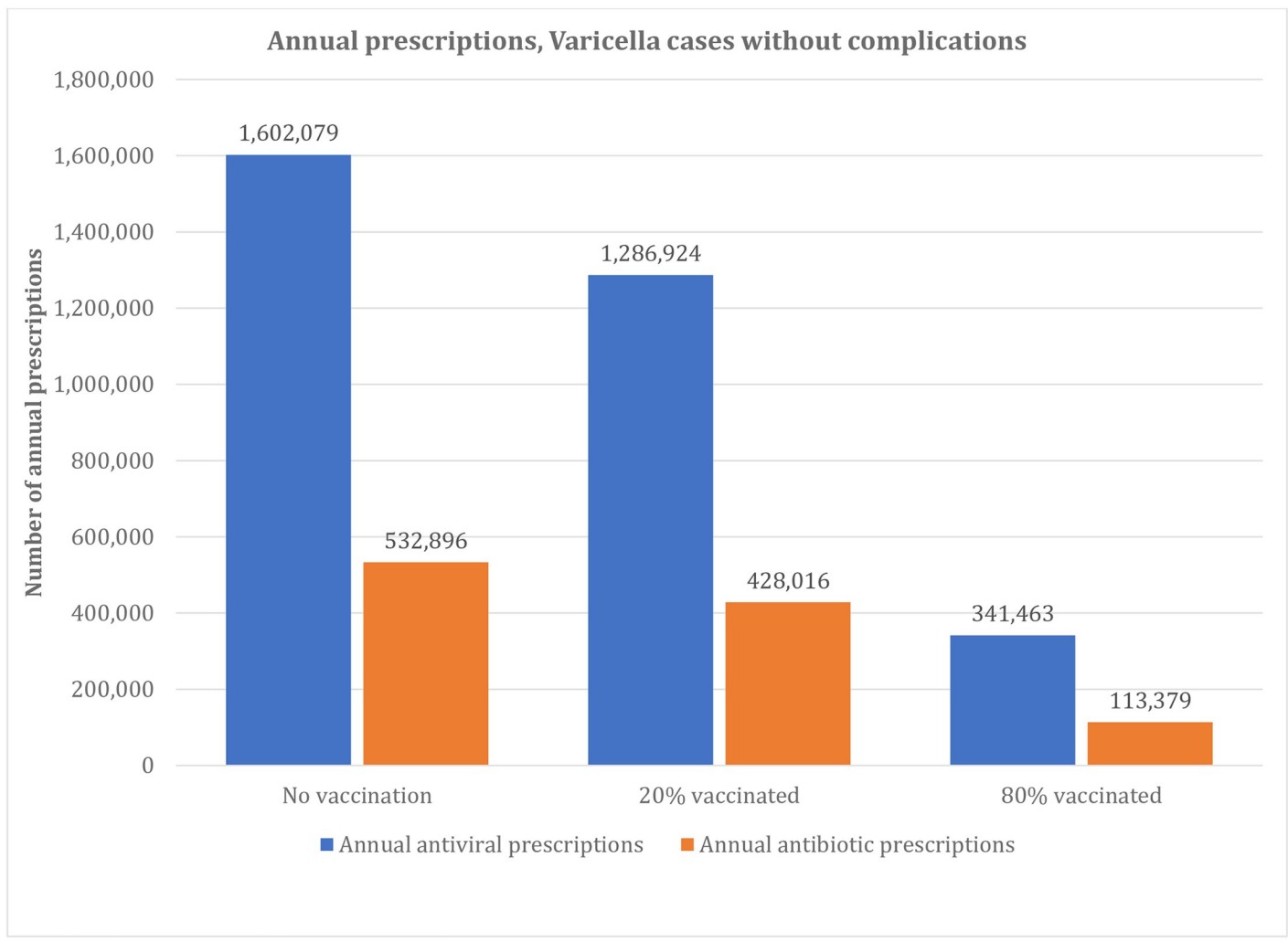

**Fig 2. Estimated annual prescriptions (varicella cases with no complications), scenario analysis.** *Total number of annual prescriptions. †Annual prescriptions avoided when compared to no vaccination scenario.

## Discussion

This study assessed the impact of UVV on the annual use of antibiotics and antivirals in the US where UVV has been implemented for over 25 years with high vaccination coverage rates. Our model demonstrated significant reductions in the annual use of antibiotics, antivirals and associated costs for varicella management, after the implementation of UVV in the US. Scenario analyses also demonstrated the high economic burden associated with the use of antibiotics and antivirals with no vaccination or lower coverage rates.

The Centers for Disease Control and Prevention (CDC) estimates one in three antibiotic prescriptions in the US is unnecessary [20] and may contribute to the risk of microbial antibiotic resistance [8]. Previous studies have shown the importance of both bacterial and viral vaccines in preventing antimicrobial prescriptions [8,21,22]. While our research does not attempt to evaluate antibiotic resistance directly, the overuse of antibiotics in the management of varicella may contribute to antimicrobial resistance.

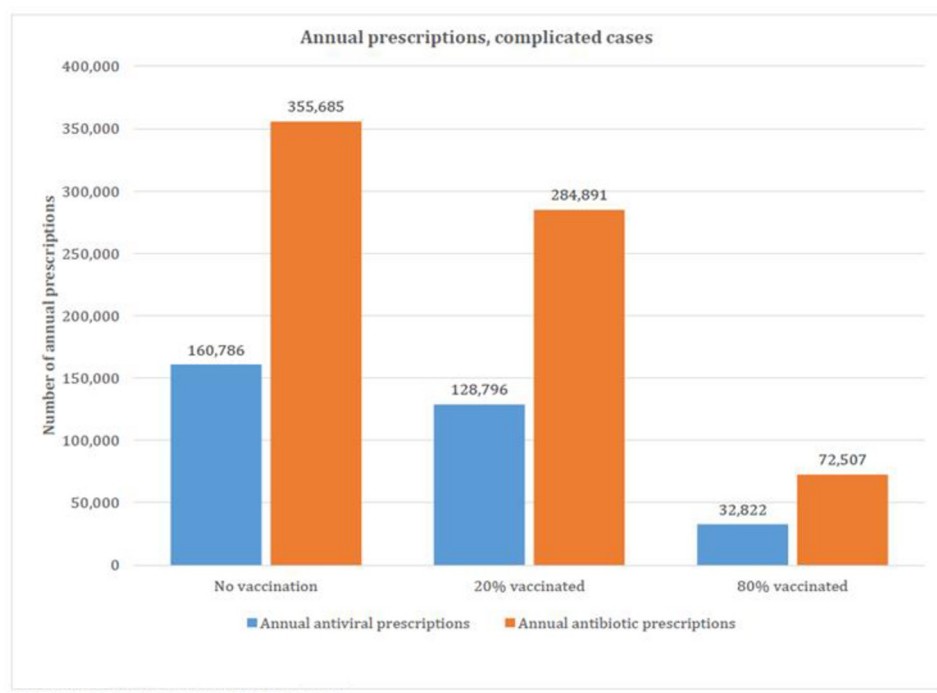

**Fig 3. Estimated annual prescriptions (cases with complications), scenario analysis.** *Total number of annual prescriptions. †Annual prescriptions avoided when compared to no vaccination scenario.

The implementation of UVV program with high vaccination coverage rates has been shown to reduce morbidity and mortality significantly, both in the US and globally [1–3,23]. Our study showed further benefits of UVV in terms of reducing prescription medication use for the management of varicella overall, and that of varicella with complications in particular, which often require antibiotics regimens for managing secondary infections. Our study estimated 30,291 varicella cases with complications annually, of which 91.4% were prescribed either antiviral or antibiotics. Our study estimated over $259 million cost saving associated with use of antibiotics and antivirals after implementation of UVV. The varicella cases with complications were not only associated with higher annual cost of antiviral and antibiotic prescriptions but also add to other direct (e.g., hospitalization, over-the-counter medications.) and indirect costs (e.g., productivity loss due to caregiver absenteeism) associated with varicella, which are not accounted for in this study [2,24–28].

Our model also showed considerable use of antibiotics and antivirals for management of varicella cases with no complications. This could be attributed to the high-risk patients as well as rarity of the disease in the US potentially leading to misdiagnosis or mistreatment of varicella cases with no complications. Use of antivirals is necessary for immunocompromised patients and older aged children [7]. Although high risk patients may need to be treated with antivirals, potential overuse of antibiotics and antivirals could be avoided by implementing strategies to improve the recognition and management of varicella infection. Since the implementation of UVV in the US, the annual number of varicella cases has substantially declined with fewer individuals seeking healthcare professional assistance. Hence, expanding education for health care providers to better recognize symptoms and manage cases with complications versus those without any complication, could further reduce the use of antibiotics, antivirals, as well as other medications such as immunoglobulin therapies.

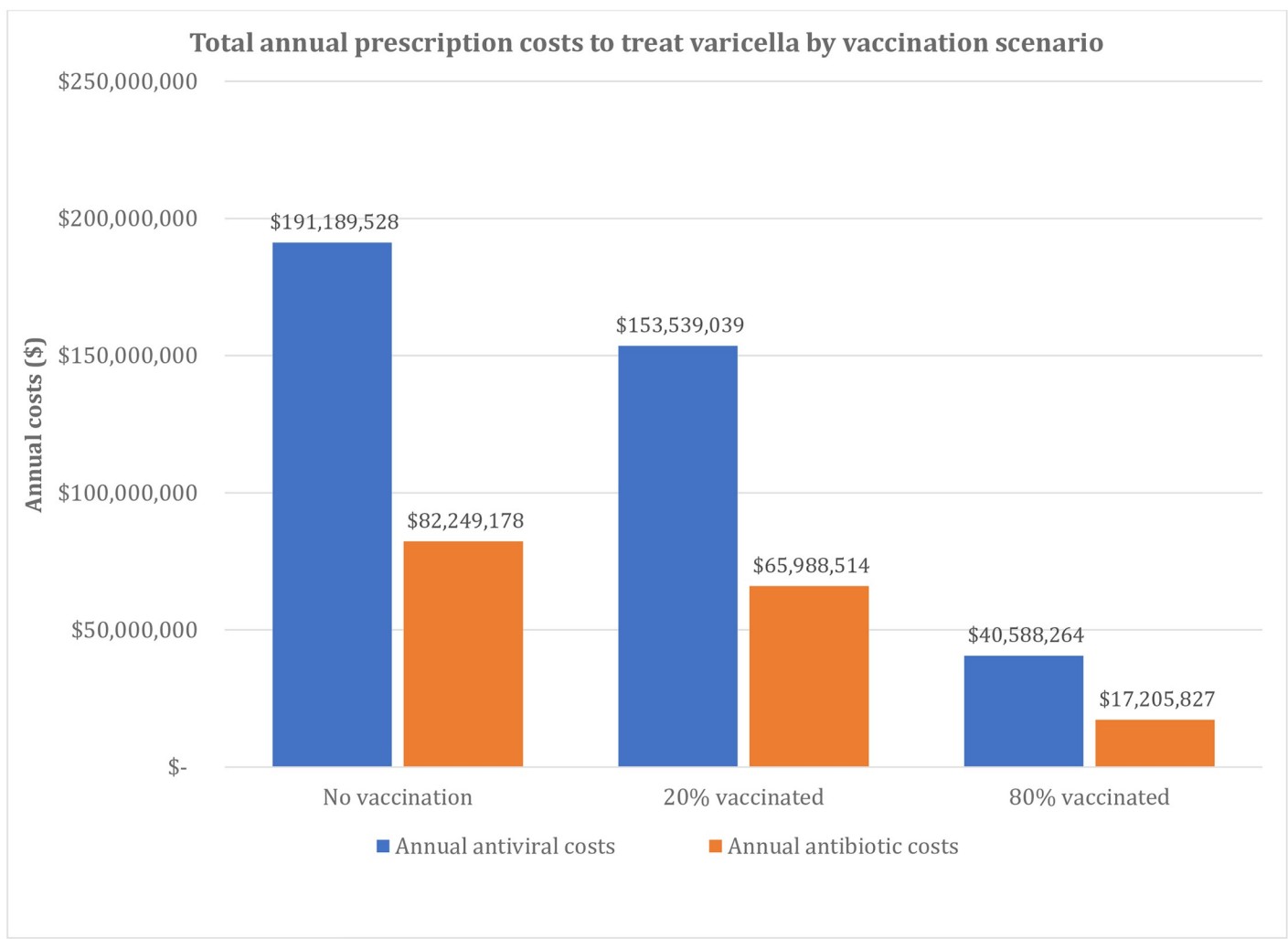

**Fig 4. Estimated annual prescription costs to treat varicella, scenario analysis.** *Total annual prescriptions costs. †Percent reduction compared to no vaccination scenario.

Our scenario analyses highlighted the importance of higher vaccination coverage after implementing UVV. The use and costs of antibiotics and antivirals were highest in the no vaccination scenario (0% coverage) and decreased with increasing vaccination coverage. Based upon the US experience, countries that have implemented or are planning to implement UVV should consider strategies to improve and maintain vaccination coverage over 80% in order to significantly reduce the use and costs of antivirals and antibiotics.

Our study has several limitations. This is a cross-section predictive model consists of only pediatric population and not an epidemiological model with entire US population. It does not account for the indirect benefits among the unvaccinated (including adults) through herd immunity effects. Hence, this model cannot provide any insights on the herd immunity with varicella vaccination. However, there is strong real world evidence supporting herd immunity due to varicella vaccination in the US with significant reductions reported in varicella incidence and varicella related hospitalizations even among the unvaccinated [10,29–32]. Although herd immunity for varicella is achieved in the US, VZV can also be transmitted to

unvaccinated children from adults with herpes zoster; however, it is less contagious than that from patients with varicella. A household study reported that 16% of susceptible children <15 years old exposed to herpes zoster developed varicella [33]. Some clinical parameters used in our model such as the annual incidence of varicella and incidence of cases with complications after 1 and 2 doses were derived from randomized control trials, which did not account for herd immunity effects. Hence, our model estimated a larger number of varicella cases annually (551,434 cases i.e. 0.75% of children under 18 years of age in 2017) than current estimates reported by the CDC (350,000 cases annually).(12) Hence, our model may have underestimated the benefits and cost-savings related to vaccination and provided more conservative estimates of benefits of vaccination.

The data on prescribing patterns came from a previously conducted survey that provided physicians with 8 patient vignettes [19]. While these were developed using literature review and expert consultation [14], they may not reflect all potential patient profiles or treatment decision-making in clinical practice accurately. Our analysis focused on only antibiotic and antiviral prescribing. Other prescriptions for immunoglobulins or the use of over-the-counter medications like antihistamines were not measured, resulting in more conservative medication cost estimates.

## Conclusion

Our model estimated substantial reductions in annual antiviral and antibiotic use and associated costs among children after implementation of a UVV program in the U.S. The model also demonstrated that increased vaccination coverage resulted in significant reductions in antibiotic and antiviral use and costs associated with varicella management. Based upon the US experience, universal varicella vaccination could be considered as a strategy to reduce the use of antibiotics and antivirals which may further help with reducing the risk of antibiotic resistance.

## Supporting information

**S1 Appendix. Additional supporting information.**
(DOCX)

## Acknowledgments

We would like to acknowledge Dr. Lara Wolfson for her contributions to the study concept and design; Joanna MacEwan and Taylor Schwartz for their contributions to the study design and data analysis; Richard Murphy for his contributions to manuscript preparation, and Shikha Surati for her project management support and data validation. Part of this research was presented at the European Society for Paediatric Infectious Diseases Virtual Meeting, May 24–29, 2020.

## Author Contributions

**Conceptualization:** Jaime Fergie, Salome Samant, Phani Veeranki, Oliver Diaz.

**Data curation:** Carolyn Harley, Salome Samant.

**Methodology:** Phani Veeranki, Oliver Diaz.

**Project administration:** Carolyn Harley.

**Supervision:** James H. Conway.

**Validation:** Jaime Fergie, James H. Conway.

**Writing – original draft:** Carolyn Harley.

**Writing – review & editing:** Manjiri Pawaskar, Jaime Fergie, Salome Samant, James H. Conway.

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
