## [Decision Letter · Decision Letter 0]

22 Mar 2022

PONE-D-21-40227IMPACT OF UNIVERSAL VARICELLA VACCINATION ON THE USE AND COST OF ANTIBIOTICS AND ANTIVIRALS FOR VARICELLA MANAGEMENT IN THE UNITED STATESPLOS ONE

Dear Dr. Pawaskar,

Thank you for submitting your manuscript to PLOS ONE. After careful consideration, we feel that it has merit but does not fully meet PLOS ONE’s publication criteria as it currently stands. Therefore, we invite you to submit a revised version of the manuscript that addresses the points raised during the review process.

Please submit your revised manuscript by May 06 2022 11:59PM If you will need more time than this to complete your revisions, please reply to this message or contact the journal office at plosone@plos.org. Please include the following items when submitting your revised manuscript:A rebuttal letter that responds to each point raised by the academic editor and reviewer(s). You should upload this letter as a separate file labeled 'Response to Reviewers'.A marked-up copy of your manuscript that highlights changes made to the original version. You should upload this as a separate file labeled 'Revised Manuscript with Track Changes'.An unmarked version of your revised paper without tracked changes. You should upload this as a separate file labeled 'Manuscript'.

We look forward to receiving your revised manuscript.

Kind regards,

Georges M.G.M. Verjans, MSc, PhD

Academic Editor

PLOS ONE

Journal Requirements:

2. PLOS requires an ORCID iD for the corresponding author in Editorial Manager on papers submitted after December 6th, 2016. Please ensure that you have an ORCID iD and that it is validated in Editorial Manager. To do this, go to ‘Update my Information’ (in the upper left-hand corner of the main menu), and click on the Fetch/Validate link next to the ORCID field. This will take you to the ORCID site and allow you to create a new iD or authenticate a pre-existing iD in Editorial Manager. Please see the following video for instructions on linking an ORCID iD to your Editorial Manager account: https://www.youtube.com/watch?v=_xcclfuvtxQ.

“”This work was funded by Merck & Co.

5. We noted in your submission details that a portion of your manuscript may have been presented or published elsewhere. [DETAILS AS NEEDED] Please clarify whether this [conference proceeding or publication] was peer-reviewed and formally published. If this work was previously peer-reviewed and published, in the cover letter please provide the reason that this work does not constitute dual publication and should be included in the current manuscript.

Additional Editor Comments (if provided):

Dear Dr Pawaskar and colleagues,

Your manuscript has been reviewed by two experts in the field. Both reviewers and I agree that your manuscript is well written and of interest to the field. Minor revisions are requested to improve the manuscript, please do so per reviewers' suggestions.

Best regards,

Georges MGM Verjans

**www.herpeslab.nl**

Dept. Viroscience

Rotterdam

The Netherlands

Reviewers' comments:

Reviewer's Responses to Questions

**Comments to the Author**

1. Is the manuscript technically sound, and do the data support the conclusions?

Reviewer #1: Yes

Reviewer #2: Yes

2. Has the statistical analysis been performed appropriately and rigorously? 

Reviewer #1: Yes

Reviewer #2: Yes

3. Have the authors made all data underlying the findings in their manuscript fully available?

Reviewer #1: Yes

Reviewer #2: Yes

4. Is the manuscript presented in an intelligible fashion and written in standard English?

Reviewer #1: Yes

Reviewer #2: Yes

5. Review Comments to the Author

Reviewer #1: This manuscript presents a version of a cost-benefit analysis for varicella vaccination in the United States in 2017. The authors show the expected cost reductions associated with universal vaccination in the USA. The goal of the report appears to be an opportunity to convince other countries to authorize universal varicella vaccination. A few comments are listed below.

1. Results, Varicella cases, lines 124-8 and Table 1. This table contains numbers that are not widely known among vaccine experts. The authors state that 93.8% of children in the USA were vaccinated against varicella in 2017. Yet the authors estimate that 396,633 cases of wild-type varicella (chickenpox) occurred in the remaining unvaccinated population of children. Please write more in the text about how the authors arrived at the number of 396,633. Furthermore, the authors estimate that 140,395 cases of break-through varicella occurred in the fully vaccinated children. Please write more in the text about how the authors arrived at the number of 140,395.

2. Add new section near the top of the Discussion about varicella herd immunity. If the number of cases of varicella in Table 1 are correct, the USA does not appear to have achieved herd immunity for varicella, even with an immunization rate of 93.8%. This is a striking conclusion, especially when we are at the end of the COVID-19 epidemic. In the USA, about 65% of the population has received complete COVID-19 vaccination. Can the authors state whether they think varicella or COVID-19 is more contagious? If they are roughly equally contagious, based on this report, the USA will never achieve herd immunity against COVID-19 by immunization, since we will never achieve greater than 92% COVID-19 vaccination. After the COVID-19 epidemic, there appears to be ever greater resistance to vaccination in the USA. Please discuss the above points in a new paragraph in the Discussion about what is meant by herd immunity and whether we have achieved herd immunity to varicella in the USA in 2017?.

3. Abstract. Suggest that one sentence about herd immunity be added into the Abstract. Was there herd immunity in the USA in 2017 against wild type varicella? Presumably herd immunity would never be achievable with an 80% immunization rate?

Reviewer #2: The aim of this study was to estimate the use of antivirals and antibiotics for treating varicella in children. The impact of vaccination was modeled after the experience in the U.S., where Varivax coverage is high. The use of antimicrobial therapy was estimated from a prescriber survey of 8 clinical vignettes. The parameters of the model were taken from published sources, although the actual reported varicella cases from the CDC were not used. This model was also used to predict the cost savings from reduced prescriptions in countries where vaccine coverage was absent, intermediate, or high. The findings presented here are interesting and compelling, although they should be taken as estimates only. These results could be used for future policy decisions on implementing varicella vaccine guidelines.

Line 127: correct typo, remove “were”.

Line 149: correct typo “d”.

Line 225: correct typo “nay”.

Supplemental Table 6 contains the key estimates of the model. However, it is difficult to find the “Total Varicella Cases” and the “Total Cases with no complications” and further down the “Total Cases with complications”. These rows are shaded gray and the font is bold. Why are these values not the top rows of the entire table? Why are they embedded in other rows that show the number of cases treated with antivirals or antibiotics? Every other row is a subset of these two key estimates, which form the base cases from the model.

In the Discussion, the point is raised that the estimate of total varicella cases may be high compared to the CDC estimates. How much higher? The CDC estimate should be included here, and the justification not to use that value should be explained. In fact, herd immunity will have a strong effect on the actual number of varicella cases in the U.S., because outbreaks of chicken pox are becoming rare. More often, varicella arises in unvaccinated children (or infants <1) who are exposed to adults with zoster. A more nuanced discussion of this situation could be added to the manuscript.

6. PLOS authors have the option to publish the peer review history of their article (what does this mean?). If published, this will include your full peer review and any attached files.

Reviewer #1: No

Reviewer #2: No

---

## [Author Response · Author response to Decision Letter 0]

5 May 2022

May 5, 2022

To,

Georges M.G.M. Verjans, MSc, PhD

Academic Editor,

PLOS ONE

 Reference: PONE-D-21-40227 Response to reviewers

Dear Dr. Verjans,

On behalf of my co-authors, we are pleased to resubmit our manuscript entitled “Impact of universal varicella vaccination on the use and cost of antibiotics and antivirals for varicella management in the United States” for publication review in Plos One.

 We appreciate the valuable comments and suggestions on our manuscript provided by the editor and reviewers. We have carefully studied each comment, provided point-by-point responses, and revised our manuscript accordingly. The line numbers provided correspond to the line numbers in the tracked changes version of the manuscript. We hope that these revisions meet your approval.

In addition, the corporate address of our company has changed, and the funding statement should be revised as “This study was sponsored by Merck Sharp & Dohme LLC, a subsidiary of Merck & Co., Inc., Rahway, NJ, USA (MSD). The funder had no role in study design, data collection and analysis, decision to publish, or preparation of the manuscript. M. Pawaskar and S. Samant, are employees of MSD and own stocks in Merck & Co., Inc., Rahway, NJ, USA. P. Veeranki, and C. Harley, are employees of PRECISIONheor, which received financial support from MSD, for the execution of this research. Though they received no payment for their work on this study, J. H. Conway reports grants and personal fees from Sanofi Pasteur, Pfizer, Merck, GSK, and Centers for Disease Control outside of the submitted work while J. Fergie reports personal fees from MSD, outside the submitted work.”

Part of this research was presented as an oral presentation at the European Society for Paediatric Infectious Diseases (ESPID) Virtual Meeting, May 24-29, 2020. However, the slides were not peer reviewed and were not published. The data was updated since then. 

Best regards,

On behalf of the co-authors,

Manjiri Pawaskar, PhD

Merck Sharp & Dohme LLC, a subsidiary of Merck & Co., Inc., Rahway, NJ, USA

email: manjiri.pawaskar@merck.com

---

## [Editor Report · Decision Letter 1]

1 Jun 2022

IMPACT OF UNIVERSAL VARICELLA VACCINATION ON THE USE AND COST OF ANTIBIOTICS AND ANTIVIRALS FOR VARICELLA MANAGEMENT IN THE UNITED STATES

PONE-D-21-40227R1

Dear Dr. Pawaskar,

We’re pleased to inform you that your manuscript has been judged scientifically suitable for publication and will be formally accepted for publication once it meets all outstanding technical requirements.

Kind regards,

Georges M.G.M. Verjans, MSc, PhD

Academic Editor

PLOS ONE

---

## [Editor Report · Acceptance letter]

3 Jun 2022

PONE-D-21-40227R1 

Impact of universal varicella vaccination on the use and cost of antibiotics and antivirals for varicella management in the United States 

Dear Dr. Pawaskar:

I'm pleased to inform you that your manuscript has been deemed suitable for publication in PLOS ONE. Congratulations! Your manuscript is now with our production department. 

Kind regards, 

on behalf of

Prof. Dr. Georges M.G.M. Verjans 

Academic Editor

PLOS ONE